# Conifer leaf wax acts as a source of secondary fatty alcohols in atmospheric aerosols

Yuhao Cui<sup>1,2</sup>, Eri Tachibana<sup>2</sup>, and Yuzo Miyazaki<sup>2</sup>

<sup>1</sup>Graduate School of Environmental Science, Hokkaido University, Sapporo, 060-0810, Japan <sup>2</sup>Institute of Low Temperature Science, Hokkaido University, Sapporo, 060-0819, Japan

Correspondence to: Yuzo Miyazaki (yuzom@lowtem.hokudai.ac.jp)

**Abstract.** Fatty alcohols (FAs) are major components of plant leaf surface lipids emitted into the atmosphere as primary biological aerosol particles (PBAPs). FAs in the atmosphere can act as ice-nucleating particles to form clouds that affect climate through radiative forcing and precipitation processes. Secondary FAs (SFAs) in plant waxes can act as tracers for PBAPs. However, the specific plant species that contribute to the atmospheric emissions of SFAs, as well as the factors controlling the SFA amount of atmospheric SFA emissions, remain poorly understood. In this study, we collected size-segregated aerosols and leaf samples from various plant species from a cool-temperate forest site in Hokkaido, northern Japan, during different seasons. *n*-nonacosan-10-ol was the most abundant SFA in the aerosols, which resided mostly in the supermicrometer size range, with the maximum concentration observed in spring. Among all plant leaves examined, *n*-nonacosan-10-ol was identified only in coniferous leaf samples. The mass of *n*-nonacosan-10-ol per leaf exhibited a seasonal trend similar to that of the aerosol SFA concentrations. Our results suggested that the amount of *n*-nonacosan-10-ol in aerosols was primarily controlled by the number of *n*-nonacosan-10-ol coniferous trees, which was determined by the phenology. Overall, our findings suggest *n*-nonacosan-10-ol can be used as a tracer compound for PBAPs originating from conifer leaf wax, which can be used to estimate the atmospheric emission flux of PBAPs on a global scale.

#### 0 1 Introduction

Primary biological aerosol particles (PBAPs) are emitted directly from natural sources associated with terrestrial and natural biota and can be an important component of organic aerosol and can act as ice nuclei (IN) to affect climate through cloud formation (e.g. Hader et al., 2014; Tobo et al., 2013; O'Sullivan et al., 2015; Lukas et al., 2020). PBAPs include microorganisms, dispersal units, fragments, and excretions of different biological organisms, such as bacteria, pollen, plant debris, and fungi (Després et al., 2012). Additionally, they also include the release of liquid and solid secretions from organisms into the atmosphere as aerosols (Després et al., 2012). Qiu et al. (2017) suggested that monolayers of *n*-alkyl alcohols with carbon numbers up to 30 can act as efficient ice nucleants, which is a common ability exhibited by long-chain fatty alcohols (FAs) (Vazquez De Vasquez et al., 2020). Moreover, FAs, known as terrestrial lipid biomarkers, can be transported over long distances originating from terrestrial sources and reaching remote oceans, which can affect the IN activity (Chen et al., 2021).

Primary FAs (PFAs) can be easily oxidized to carboxylic acids in the aqueous phase, with aldehydes being the intermediate components (Yu et al., 2005). Compared to PFAs, secondary FAs (SFAs) are chemically more stable owing to less Habstraction (Koivisto et al., 2015). Therefore, SFAs in atmospheric aerosols are expected to serve as tracers for specific sources of PBAPs.

SFAs are important wax components in higher plants. Most tubules (tube-like epicuticular wax structures) which possess microscopic structures visible on leaf surfaces, belong to a group of secondary alcohol tubules and predominantly comprise nonacosanol and its homologues (Huth et al., 2021). The Aerosol Characterization Experiments-Asia campaign detected two SFAs, namely n-nonacosan-10-ol and n-nonacosan-5,10-ol, in ambient aerosols at ground sites in the western North Pacific (Simoneit et al., 2004). Simoneit et al. (2004) indicated that SFA compounds in aerosols are an input of plant waxes from forests, which comprise both soft and hardwoods. Miyazaki et al. (2019) identified five SFAs in atmospheric aerosols at two forest sites in Japan and reported, for the first time, that the mass concentrations of these SFAs exhibit distinct seasonal variations. They showed that SFAs exhibit pronounced peaks during the growing season at each forest site, indicating that SFAs in aerosols primarily originate from plant waxes and can be useful tracers for PBAPs. Cui et al. (2023) evaluated the mass size distributions of five SFAs in atmospheric aerosols at a forest site and reported the peaks shift from larger size (> 10 μm) in spring to smaller size ranges (1.9–3 μm) in autumn. They suggested that the senescence status of plants affects the aerosol particle size of SFAs. Gagosian et al. (1987) emphasized that aerosol lipids can be transported from the land surface to the ocean over a relatively short timescale of a few days. This implies that the influence of aerosol SFAs is not limited to local atmospheric environments, but also on regional scales (e.g., IN activity). Overall, previous studies have suggested that SFAs in plant leaves can be emitted into the atmosphere as PBAPs, potentially influencing cloud formation through IN at a regional scale.

Although previous studies have suggested that plant waxes are the primary source of aerosol SFAs, the specific plant species contributing to the atmospheric emissions of SFAs and the factors controlling the emission processes are not well understood. In this study, we aimed to elucidate the specific plant species acting as a source of SFAs in aerosols based on simultaneous sampling of size-segregated aerosols and leaves of different plant species at a cool-temperate forest site. Additionally, we also evaluated the influence of meteorological factors, such as local wind speed and precipitation, to better understand the atmospheric emission process of SFAs from plant leaves.

# 2 Experimental Design and Methodology

Ambient aerosol and plant leaf samples were collected from the Tomakomai (TMK) experimental forest site (42°43′ N, 141°36′ E) in the western part of Hokkaido, the northernmost island of Japan (Hiura, 2001). The experimental forest spans 2715 ha and is located near the city centre of Tomakomai, which faces the Pacific Ocean. The study site is located approximately 60 km south of Sapporo (**Fig. 1**). The experimental forest site comprises a mixed cool-temperate forest consisting of mature and secondary deciduous trees and man-made coniferous trees with various types of forest floor cover (Hiura, 2001, 2005). The

dominant tree species were *Quercus crispula* (Mongolian oak), *Acer pictum* (mono maple), the human-planted species *Larix leptolepsis* (Japanese larch), *Abies sachalinensis* (Sakhalin fir), and *Picea glehnii* (Sakhalin spruce). The forest floor is mainly covered by *Dryopteris crassirhizoma* and *Parasenecio kamtschaticus*. The average height of the dominant tree species in the forest ranged from 15 m to 20 m, and the leaf area index ranged from 3.3–4.9 m<sup>2</sup> m<sup>-2</sup>.

Monthly average temperatures in the TMK experimental forest range from -3.2°C to 19.1°C, and the average annual precipitation is 1450 mm (Hiura, 2001). The snow cover reaches a maximum depth of 50 cm between December and March. The predominant local wind direction in autumn and winter is from the north, which is covered by forested areas. In contrast, air masses are typically transported from the south (from the oceanic region passing over coastal urban areas) during summer (Miyazaki et al., 2019).

Figure 1: Location of the study site where aerosol and plant leaves were sampled (©copyright by Google Map).

### 2.1 Size-segregated aerosol sampling

85

Size-segregated aerosol samples were collected on prebaked quartz fibre filters using a high-volume air sampler (Model 120SL; KIMOTO Electric Co., LTD, Osaka, Japan) attached to a cascade impactor (Model TE-234; Tisch Environmental, Cleves, OH, USA). The aerosol sample was collected in five stages at a flow rate of 1130 L min<sup>-1</sup>, without controlling temperature humidity. Aerosol particles with aerodynamic diameter ( $D_p$ ) < 0.95  $\mu$ m were collected at the bottom stage of the impactor (hereafter submicrometer particles). In contrast, supermicrometer particles ( $D_p$  > 0.95  $\mu$ m) were collected from the upper four stages of the impactor. The mass concentration of the total suspended particles (TSP) was defined as the sum of aerosol particles collected across all five stages of the impactor. The duration of each sampling was approximately one week (Miyazaki

et al., 2019; Cui et al., 2023). Consequently, the present study emphasizes data from these two seasons. At this forest site, aerosol sampling was conducted from 2–23 May (spring) in 2024, 1–8 August (summer) in 2023, and 3–24 October (autumn) in 2023.

## 2.2 Leaf sampling of different plant species

Plant leaf samples were systematically collected at the beginning or end of each aerosol sampling at a frequency of approximately once every 1–2 weeks. The leaves were sampled from three broad-leaved deciduous tree species, which included *Q. crispula* (Mongolian oak), *Acer pictum* (mono maple), and *Acer palmatum* (Japanese maple), and two coniferous evergreen tree species, which included *Abies sachalinensis* (Sakhalin fir), and *Picea glehnii* (Sakhalin spruce). Additionally, leaf samples were obtained from two ground-cover plant species, *D. crassirhizoma* and *Parasenecio kamtschaticus*. These species were selected because they dominate study site in the forest.

Broad leaves were randomly collected from different branches of the tree. As conifer leaves at different growth stages coexist in the same individual tree (Liu et al., 2020), whole branches with different leaf stages were collected during conifer leaf sampling. The plant leaf samples were stored at 4 °C in a refrigerated room prior to analysis. In this study, the measurement results of SFAs in the leaf samples are shown for samples collected in spring (9, 16, and 23 May in 2024), summer (1 August 2023), autumn (3, 17, and 31 October in 2023), and winter (30 January 2024).

# 110 2.3 Measurement of SFAs and sugar compounds in aerosol samples

To measure SFAs and related compounds, a filter cut (3.8 cm<sup>2</sup>) of the bottom stage of the impactor was used for the analysis of the submicrometer particles, whereas 8.8 cm<sup>2</sup> of the filter cut from each of the upper four stages was analyzed for the supermicrometer particles. Each filter sample was extracted using a mixture of dichloromethane (DCM) and methanol (MeOH) (2:1, v/v). The hydroxyl functional groups (-OH) in the extracted samples were derivatized using a mixture of 50 μL N,O-bis-(trimethylsilyl) trifluoroacetamide (BSTFA) and 10 µL pyridine to form trimethylsilyl (TMS) ether (-OTMS) (Fu et al., 2009). For each sample, 2 µL of the TMS derivative was injected into a capillary gas chromatograph (GC8890, Agilent Technologies) equipped with a fused silica capillary column (DB-5MS) and coupled to a mass spectrometer (MSD5977B, Agilent Technologies). The GC-MS analysis was performed in the splitless injection mode to minimize volatilization loss of longchain SFAs. The total ion chromatograms (TICs) of the five derivatized SFAs were identified, and the peak was identified based on detailed interpretation of the EI mass spectral data (mass spectrometric fragmentation patterns) together with their comparison to data in the literature as well as exact mass measurements using a high-resolution gas chromatograph-time-offlight mass spectrometer (GC-TOF-MS; JMS-T100GCV, JEOL), as described by Miyazaki et al. (2019). Quantification of the mass was performed using 140  $\mu$ l of internal standard (1.43 ng  $\mu$ L<sup>-1</sup> of C<sub>13</sub> in hexane), which was added to the derivatized sample just before the injection into the GC-MS. The concentrations of SFAs were determined by the MS peak area of TMS derivative of SFA relative to that of the internal standard. Figure 2 presents the molecular structures of the five SFAs identified in the present study.

150

155

Figure 2: Molecular structures of secondary fatty alcohols (SFAs) identified in this study: (a) *n*-nonacosan-10-ol (C29), (b) *n*-nonacosan-5,10-diol (C29), (c) *n*-nonacosan-10,13-diol (C29), (d) *n*-heptacosan-10-ol (C27), and (e) *n*-heptacosan-5,10-diol (C27).

#### 140 2.4 Bulk water-insoluble organic carbon (WIOC) in aerosol samples

To derive mass concentrations of WIOC, we measured the mass concentrations of organic carbon (OC) and water-soluble organic carbon (WSOC). The mass concentration of OC was measured using a Sunset Lab OC/EC analyzer. To measure WSOC concentration, a filter cut was extracted with ultrapure water and filtered with a disc filter (Millex-GV, 0.22 μm, Millipore, Billerica, MA, USA), followed by injection of the extracts into a total organic carbon analyser (Model TOC-LCHP, Shimadzu) The OC and WSOC concentrations were calculated using filter blanks. The mass concentration of WIOC was defined as the difference between that of OC and WSOC (i.e., [WIOC] = [OC] – [WSOC]).

# 2.5 Measurements of SFAs in plant leaf samples

All leaf samples were dried in a desiccator at room temperature. Broadleaf samples were ground and homogenized in a mortar and coniferous leaf samples were analyzed without grinding. In general, SFAs in plant waxes exist in both free and esterified forms. Free-form SFAs were extracted from individual leaves of each species using a mixture of DCM and MeOH (2:1, v/v) in an ultrasonication bath, and the extracts were filtered through quartz wool. To quantify the total mass of SFAs (free + esterified forms), the extracts underwent alkaline hydrolysis procedure (Angst et al., 2017) with methanolic KOH at 80 °C for 3 h to measure esterified forms of SFA mass in addition to the free forms. Ultrasonication was applied for the first 15 min of each hour to prevent excessive heating. After cooling, the hydrolysates were re-extracted with DCM and MeOH (2:1, v/v) and subsequently combined with KOH extracts. The combined solution was filtered and acidified to pH1 using 30% (w/w) concentrated hydrochloric acid (HCl). The acid fractions were separated from the extracts by liquid-liquid extraction. The solution in the organic layer was evaporated and concentrated using a rotary evaporator and then dried under N<sub>2</sub>. Once the solvent was completely removed, the final extracts of both the free (without hydrolysis) and esterified (with hydrolysis) forms

were derivatized using BSTFA and pyridine (5:1, v/v). The derivatization and subsequent identification procedures were the same as those used for the analysis of free-form SFAs in aerosols. The extracted samples were injected into the GC–MS using the same procedure as that used for the aerosol samples. In this study, the SFA mass per leaf, both in free and total (free + esterified) forms, was derived.

#### 3 Results and Discussion

#### 3.1 Seasonal differences in the mass concentrations and mass size distributions of SFAs in aerosol samples

Figure 3 shows the typical mass size distributions of the five SFAs in the aerosol samples collected at the Tomakomai (TMK) experimental forest site from 2–9 May, 2024. n-nonacosan-10-ol was the most abundant SFAs identified in all samples, followed by n-nonacosan-5,10-diol. The majority of the SFAs mass resided in the supermicrometer (D<sub>p</sub> > 0.95 μm) size range. The dominance of n-nonacosan-10-ol with the majority of the mass in the supermicrometer size was observed in all the aerosol samples, which has been also reported for other forest sites (Miyazaki et al., 2019; Cui et al., 2023). In this study, n-nonacosan-10-ol is mainly discussed as a representative compound of SFAs.

Figure 3: Mass size distributions of secondary fatty alcohols (SFAs) recorded between 2 May and 9 May in 2024.

**Figure 4** shows the mass concentrations and fractions of *n*-nonacosan-10-ol in submicrometer particles and TSP. The concentration of *n*-nonacosan-10-ol in TSP was the highest in spring (180.3±38.1 ng m<sup>-3</sup>), followed by autumn (13.6±11.3 ng m<sup>-3</sup>) and summer (4.73 ng m<sup>-3</sup>). The maximum concentration of *n*-nonacosan-10-ol in spring has also been observed at other temperate and cool-temperate forest sites (Miyazaki et al., 2019; Cui et al., 2023). The average concentrations of *n*-nonacosan-10-ol in the TSP in spring were similar to those reported for the same forest site in a different year (Miyazaki et al., 2019). In contrast, the observed mass concentrations of *n*-nonacosan-10-ol in spring were approximately 20 times higher

than those observed at the Sapporo Forest Meteorology Research (SPR) site, which is a mixed cool-temperate forest consisting of deciduous trees (Cui et al., 2023). As shown above, the majority of the SFA mass resided in the supermicrometer size range across all seasons. Specifically, the mass of *n*-nonacosan-10-ol in the supermicrometer size range accounted for 77.7%, 67.5%, and 82.0% of the total mass in spring, summer, and autumn, respectively.

200

215

220

205 Figure 4: Mass size concentrations of *n*-nonacosan-10-ol in submicrometer particles (Sub-μm P; white bars) and total suspended particulate matter (TSP; gray bars) collected at the TMK site in spring (2–23 May 2024), summer (1–8 August 2023), and autumn (3–24 October 2023). For spring and autumn, each value represents the average of three sample sets weekly obtained. In summer, the value of the single sample set is shown because only one sample was obtained in that season. The open circles show the mass ratios of sub-μm P to TSP. Note that the left y-axis is shown on a logarithmic scale to show the large variation of the mass concentrations in each particle size range.

Figure 5 presents the mass size distributions of *n*-nonacosan-10-ol in the aerosols during each season. In the spring, the peak diameter of *n*-nonacosan-10-ol was larger than 7.2 μm. On the other hand, the peak of *n*-nonacosan-10-ol resided in the smaller particle size range of 1.5–3.0 μm in autumn. Cui et al. (2023) reported a similar seasonal trend in the mass size distribution of SFAs in aerosols at another cool-temperate forest site (SPR site). They also showed that the peak of the SFA mass size range was in a similar range to that of the supermicrometer particles in spring, whereas the relative contribution of the mass in the smaller size ranges in the supermicrometer particles became larger in autumn. Their results suggested that SFAs originate mostly from plant waxes and that the leaf senescence status is likely an important factor controlling the size distribution of SFAs. As the seasonal change in the size distributions of aerosol SFA in this study is similar to that observed by Cui et al. (2023), the current results suggest that plant leaf wax is a possible source of aerosol SFA.

235

Figure 5: The mass size distributions of *n*-nonacosan-10-ol in aerosols obtained in (a) spring (means of multiple samples collected during 2–23 May 2024), (b) summer (a single sample collected during 1–8 August 2023), and (c) autumn (means of multiple samples collected during 3–24 October 2023).

## 3.2 *n*-nonacosan-10-ol in plant leaf samples

To elucidate the contributions of plant wax to atmospheric aerosols and the specific plant species acting as sources of SFAs in atmospheric aerosols, SFAs in the leaf samples were examined for all plant leaves collected at the study site. As shown in **Table 1**, among the leaf samples collected from all plant species, *n*-nonacosan-10-ol was only identified in the leaves of *Abies sachalinensis* (Sakhalin fir) and *Picea glehnii* (Sakhalin spruce). *n*-nonacosan-10-ol was not detected in the broadleaf samples examined in this study. Sakhalin fir was the most widely distributed conifer tree species in the study area, but its sample could not be obtained in winter because the area was inaccessible owing to snow accumulation during that season.

Table 1: Detection of secondary fatty alcohols (SFAs) in the seven plant species sampled from the study site. ND indicates that a compound was not detected.

| Plant Type         | Scientific Name           | Common Name             | Detection of SFAs |
|--------------------|---------------------------|-------------------------|-------------------|
| Broad-leaved tree  | Quercus crispula          | Mongolian oak           | ND                |
|                    | Acer palmatum             | Japanese maple          | ND                |
|                    | Acer pictum               | Mono maple              | ND                |
| Coniferous tree    | Abies sachalinensis       | Sakhalin fir            | Detected          |
|                    | Picea glehnii             | Sakhalin spruce         | Detected          |
| Ground cover plant | Dryopteris crassirhizoma  | Thick stemmed wood fern | ND                |
|                    | Parasenecio kamtschaticus | (Herbaceous species)    | ND                |

Because SFAs in plant leaf waxes typically exist in both free and esterified forms, **Figure 6** compares the SFA mass per conifer leaf in the free form with that in all (free and esterified) forms. The mass of SFAs in the free form  $(69.2 \pm 7.32 \,\mu\text{g})$  was substantially larger than that in the esterified form  $(2.17 \pm 1.23 \,\mu\text{g})$ , showing that *n*-nonacosan-10-ol in the conifer leaf samples was predominantly in the free form. The dominance of the free form of *n*-nonacosan-10-ol in the conferring leaves is also supported by previous studies (Kolattukudy, 1976; Kolattukudy et al., 1989). Therefore, it is reasonable to compare free-form *n*-nonacosan-10-ol in aerosols with that in conifer leaves.

245

240

250

255 Figure 6: Mass of *n*-nonacosan-10-ol per conifer (*Abies sachalinensis*) leaf in the free and all (free + esterified) forms from samples collected on 9 May 2024. Bars indicate the standard deviation of the average mass of three replicate leaves collected from similar positions on a conifer tree.

#### 3.3 Seasonal variations in the mass of *n*-nonacosan-10-ol in conifer leaves

We investigated the seasonal variations in the mass of *n*-nonacosan-10-ol in conifer (Sakhalin fir) leaves to assess the possible link between the plant wax of conifer leaves and atmospheric aerosols in terms of the atmospheric emission of SFAs. The positions of the plant leaves on the branches defined in this study are shown in **Fig. S1**. The most abundant leaf type among the conifer species sampled in this study was located in the middle of the branches (**Fig. S1**). **Figure 7** shows the mass of *n*-nonacosan-10-ol per needle leaf in the middle of the branch during each season. The average mass of *n*-nonacosan-10-ol was largest in spring (62.1±15.0 μg), followed by autumn (58.3 ± 5.06 μg) and summer (16.2 ± 1.66 μg), although the difference in the mass between spring and autumn was insignificant. The seasonal trend of the *n*-nonacosan-10-ol mass in the conifer leaf samples was similar to that of the mass concentration in the aerosols (**Fig. 4**), suggesting a potential role for *n*-nonacosan-10-ol in linking plant phenology with atmospheric aerosol chemical composition.

Figure 7: Mass of *n*-nonacosan-10-ol per confer leaf at the study site in each season. Note that the data of conifer leaves were taken from Sakhalin fir (*Abies sachalinensis*) for spring, summer and autumn, whereas Sakhalin spruce (*Picea glehnii*) for winter due to an issue of accessibility to the sampling location. Bars indicate the standard deviation of the average mass in leaf samples measured in each season, with all leaves collected from the middle part of branches. Detailed sample information is provided in Table S4.

A similar seasonal trend of the *n*-nonacosan-10-ol mass in plant leaves was reported for *Pinus pinaster* (Nikolić et al., 2020), where the mass fraction of n-nonacosan-10-ol to leaf cuticular wax in needle leaves was slightly higher in spring (79.0%) than in autumn (75.2%). Jenks et al. (2002) suggested that the amount of plant leaf wax varies depending on the developmental stage of the plant leaf. In the present study, the largest mass of n-nonacosan-10-ol in the leaf samples was observed in spring, which could be attributed to the active generation and regeneration of plant waxes during the growing season (Mohammadian et al., 2007). Interactions between phylogeny and meteorological parameters (e.g., temperature and precipitation) can influence plant physiological characteristics (Wang et al., 2018). The conifer leaves sampled in summer were older than those sampled in spring, and their surface waxes were substantially degraded by the local wind and precipitation. Moreover, the capacity for wax synthesis is likely reduced by heat stress (Jenks et al., 2002). In contrast, the increase in SFA mass in the leaf samples in autumn can be explained by the hydrophobicity of the waxes in that season to protect plants against water loss (Kreyling et al., 2012). Plants protect themselves from water loss by increasing the turnover of wax components towards hydrophobic aliphatic compounds under cold stress, which is supported by the findings of previous studies on conifer species (Cape and Percy, 1993; Shepherd and Wynne Griffiths, 2006; Kreyling et al., 2012). In winter, the mass of n-nonacosan-10-ol per one conifer leaf  $(2.32 \pm 0.34 \,\mu\text{g})$  was substantially lower than those in the other season, which may be caused by natural wax erosion (Simões et al., 2022) and slow plant growth (Caffarra and Donnelly, 2011) in the cold season. These phenological development processes are controlled by environmental factors such as shorter illumination times and limited availability of water in the soil caused by freezing conditions in winter.

The tubular wax crystals contain SFAs usually ranging from approximately 0.5 to 3 µm (Koch et al., 2009). Wax particles remain in older leaves where the size of the wax tends to be smaller than the maximum size range (Tomaszewski, 2004). In other words, the wax production in newly formed leaves is accompanied by both vertical and horizontal expansion, with the

maximum diameter of wax structures reaching 11-14 µm as plant leaves develop. This size range corresponded to mass size range of the SFAs in the aerosols observed in the present study (Fig. 5). Seasonal differences in the size distributions of SFAs in aerosols shown in Fig. 5 suggests that the size distribution was closely linked with the structural characteristics of leaf waxes. Seasonal variations in SFA mass observed in both atmospheric aerosols and plant leaves suggested that plant phenology likely influenced SFA emissions. As a representative SFA, n-nonacosan-10-ol is not only a major component of Sakhalin fir leaf waxes but is also abundant in waxes produced by several plant species (e.g. Dommisse et al., 2007; Zhang et al., 2024), including Pinus halepensis (58% of total leaf epicuticular waxes; Matas et al., 2003), Tropaeolum majus (67% of total leaf waxes; Koch et al., 2006), and Pinus pinaster (77% of total leaf cuticular waxes; Nikolić et al., 2020). SFAs commonly exist in many plant species and are formed during wax biosynthesis in leaves. The wax biosynthesis occurs via three distinct and parallel biosynthetic pathways; (i) the decarbonylation, (ii) the acyl reduction, and (iii) β-ketoacyl-elongation. This decarbonylation pathway leads to the formation of aldehydes, odd-chain alkanes, SFAs, and ketones. In contrast, the acylreduction pathway produces primary alcohols and esters. The  $\beta$ -ketoacyl elongation pathway results in the synthesis of  $\beta$ diketones and their derivatives (Post-Beittenmiller, 1996). Among SFAs, n-nonacosan-10-ol is the major component of tubular plant wax crystals (Dommisse et al., 2007), which is consistent with observations made for leaf samples in the present study. However, the detailed SFA biosynthetic pathways in plant leaves remain unclear. Zhang et al. (2024) proposed a possible mechanism that n-nonacosan-10-ol can be formed C<sub>12</sub> or C<sub>22</sub> elongation intermediates with 3-ketoacyl or 3-hydroxyacyl structures. The presence of SFAs in plant leaf wax serves as the foundation for multiple hypotheses regarding their biosynthesis and provides insights into the potential formation process of SFAs in atmospheric aerosols. Although the biosynthetic mechanisms of SFAs in plant leaves are not fully understood, information about SFA distribution along branches may provide additional insights into their formation. As a new leaf appears at the tip of a conifer branch, leaves at different positions on the branch represent their growth stages, thereby providing information on how leaf age affects wax accumulation.

**Figure 8** shows the mass of *n*-nonacosan-10-ol in conifer leaves at the middle and top positions of a branch obtained in spring and autumn. In both seasons, the mass of *n*-nonacosan-10-ol in the leaves in the middle position was approximately twice that of leaves obtained from the top. This difference can be explained by wax accumulation synchronizing with leaf development, which is more pronounced in leaves at the middle position (Busta et al., 2017). The following section further examines how leaf senescence affects SFA mass in conifer leaves.

Figure 8: The mass of *n*-nonacosan-10-ol per Sakhalin fir (*Abies sachalinensis*) leaf in different position of one branch in spring (25 May 2023) and autumn (3 Oct 2023). Bars indicate the standard deviation of the average mass of three coniferous leaf samples.

## 3.4 Dependence of the *n*-nonacosan-10-ol mass on senescent status of conifer leaves

We investigated the dependence of SFA mass on the senescent status of conifer leaves. **Figure 9** compares the mass of *n*-nonacosan-10-ol per leaf at three different positions on one branch obtained on 3 October 2023. In addition to top and middle positions, "brown" parts of the conifer leaves on a branch are defined in **Fig. S1** in the Supplementary Material. Brown leaves occupy a minor fraction of the total conifer leaves and are the most senescent among all parts of conifer leaves. Brown leaves (140 µg) accounted for the highest fraction in the total mass of *n*-nonacosan-10-ol, followed by the green mature leaves at middle positions (67.1 µg). Brown leaves exist on branches for a long time after senescence, with complete resorption processes (Yuan et al., 2018). The large mass of SFAs (i.e., leaf wax) in brown leaves can be explained by the continued activity of wax biosynthetic genes during leaf senescence (Laila et al., 2017), followed by the accumulation of plant wax from the inner to the outer leaves in senesced plant leaves. This continued activity may function as a protective mechanism against certain plant species; however, the exact mechanism remains largely unknown. Our results indicated that SFA mass in conifer leaves depends on the plant growth stage and senescence status.

Figure 9: The mass of *n*-nonacosan-10-ol per Sakhalin fir (*Abies sachalinensis*) leaf in different positions of one branch in autumn (3 Oct 2023). Bars indicate the standard deviation of the average mass from three coniferous leaves.

# 3.5 Possible key process in the emission of aerosol SFAs

The possible factors controlling SFAs concentrations in ambient aerosols at the study site were examined to determine the atmospheric emission processes of SFAs from conifer leaves. **Figure 10** compares the average mass concentrations of *n*-nonacosan-10-ol in aerosols with those of *n*-nonacosan-10-ol in *Abies sachalinensis* (Sakhalin fir) leaves. The figure also shows the average wind speeds and accumulated precipitation in spring and autumn, when the mass of *n*-nonacosan-10-ol in both aerosols and conifer leaves showed large values.

In spring, the temporal trend of *n*-nonacosan-10-ol mass concentrations in aerosols was similar to that of the local wind speeds and *n*-nonacosan-10-ol mass in the leaves. In contrast, the relationship between aerosols and accumulated precipitation was not clear. Local winds can typically blow off plant wax from the leaf surface into the atmosphere, which is a common atmospheric emission process for PBAPs (Tegen and Schepanski, 2018). At higher wind speeds, plant waxes can be removed from leaves through mechanical processes caused by aerodynamic forces and leaf-to-leaf contact, such as crystal fracturing and abrasion (Shepherd and Wynne Griffiths, 2006). Our findings suggested that local wind speeds and the amount of wax on conifer leaves are important factors that control the mass concentrations of SFAs in aerosols in spring.

405

395

Figure 10: (a, b) Mass concentration of *n*-nonacosan-10-ol in TSP, (c, d) average local wind speeds, (e, f) accumulated precipitation, and (g, h) the mass of *n*-nonacosan-10-ol in the conifer leaves during each period/day in spring (a, c, e, g) and autumn (b, d, f, h). In (f), accumulated precipitation during Oct 10–17 was zero.

Precipitation can remove SFAs, both atmospheric aerosols and plant wax, from leaf surfaces. In spring, the size of aerosol SFAs was predominantly coincided with the supermicrometer range (**Fig. 5**) and can be removed more efficiently by precipitation than submicrometer particles (Rathnayake et al., 2017). Even after removal by precipitation, the concentration level of PBAPs often recovers within hours to days after rainfall ceases (e.g., Guo et al., 2016), which occurred on a timescale of within one week of aerosol sampling in the present study. Although it was difficult to systematically explain the effect of precipitation on the temporal trend of aerosol SFA concentration in this study, the SFA mass in conifer leaves, combined with local wind speeds, likely increased the atmospheric aerosol concentrations of SFA in spring.

The seasonal difference in the SFA mass concentration in aerosols between summer and autumn was more significant than the difference in the SFA mass in conifer leaves. The relationships among aerosol SFA concentrations, local wind speeds,

precipitation, and SFA mass in leaves were not clear in autumn. In spring, larger plant wax particles may be more easily removed from the leaf surface and emitted into the atmosphere through the physical processes described above. On the other hand, plant wax in autumn is generally smaller than that in spring and can be removed less from the leaves into the atmosphere. These processes can partly result in seasonal differences in atmospheric SFA emissions from conifer leaves. This interpretation was supported by the observed mass size distribution of aerosol SFA as larger particle size range ( $D_p > 7.2 \mu m$ ) peaked in spring, whereas smaller size range ( $D_p = 1.5-3.0 \mu m$ ) peaked in autumn (Fig. 5). However, the formation mechanism of tubules (a type of plant wax structure in which SFAs are the major constituents) remains poorly understood (Huth et al., 2021). Therefore, further research is required to elucidate these underlying processes.

We further investigated the influence of site-specific factors (e.g., abundance and differences in coniferous species) on aerosol SFA concentration levels. Figure 11 compares the average mass concentrations of n-nonacosan-10-ol in bulk aerosols (in all 430 size ranges) reported for different forest sites during spring. The average mass concentration of *n*-nonacosan-10-ol in spring is greatest at Fuji-Hokuroku (FHK) flux research site (212±232 ng m<sup>-3</sup>; Miyazaki et al., 2019), followed by TMK site (180±38.1 ng m<sup>-3</sup>; the present study study). Among the three sites, the average concentration is significantly lower at SPR site (10.2±10.3 ng m<sup>-3</sup>) (Cui et al., 2023). The differences in the concentrations of aerosol SFAs at the three forest sites can be attributed to differences in the types and abundance of coniferous tree species. The predominant tree species at the FHK site are conifers, 435 including Japanese larch (Larix kaempferi Sarg.) interspersed with evergreen needle-leaf species (Pinus densiflora and Abies homolepis) (Takahashi et al., 2015). The TMK site consists of mature and secondary deciduous trees, together with coniferous trees planted by humans, where coniferous trees are not the dominant species. The SPR site is mainly covered with broadleaf trees, with fewer coniferous trees than the FHK and TMK sites (Nakai et al., 2003). At those three forest sites, the average local wind speeds observed during each sampling period at each site were in the similar range of 2.4–3.0 m s<sup>-1</sup> during each 440 aerosol sampling period. Therefore, the differences in the mass concentrations of n-nonacosan-10-ol in aerosols between the conifer-rich sites (TMK and FHK) and the broadleaf-dominated site (SPR) indicate that conifer leaves are the primary source of aerosol SFAs.

455

Figure 11: The average mass concentrations of *n*-nonacosan-10-ol in aerosols reported for three different forest sites in spring. Each aerosol samples were obtained at Sapporo (SPR) Forest Meteorology Research Site (Cui et al., 2023), Tomakomai (TMK) experimental forest, and Fuji-Hokuroku (FHK) flux research site (Miyazaki et al., 2019). For details of each forest site, see text.

We found that SFAs can be used as tracers for PBAPs originating from coniferous leaf waxes. Cui et al. (2023) suggested that atmospheric emissions of SFAs from plant leaf wax could be an important factor controlling the aerosol mass concentration of WIOC in spring (i.e., growing season). In this study, the average mass concentration ratio of n-nonacosan-10-ol to WIOC was  $0.11\pm0.02$  (ng ng<sup>-1</sup>) in spring. Notably, the aerosol mass size distributions of n-nonacosan-10-ol and WIOC were similar in the spring (**Fig. S2a** in Supplementary Material). A significant linear correlation was observed between their mass concentrations during that season, with a coefficient of determination ( $R^2$ ) of 0.90 (**Fig. S2b** in Supplementary Information), indicating that plant leaf wax is an important factor controlling the aerosol mass concentration of bulk WIOC in spring. These findings further support the use of SFAs as effective tracers of PBAPs originating from coniferous leaf waxes.

At the TMK site, the aboveground biomass (AGB) of the conifer plantations accounted for approximately 43% of the total AGB (Hiura, 2005). Coniferous trees are the dominant species in boreal forests and are major components of temperate forests, where boreal and temperate forests account for 32% and 14% of global forest cover, respectively (Pan et al., 2011). The results of this study, together with those of previous studies on aerosol SFAs, suggest that coniferous leaf wax is an important source of PBAPs, particularly in spring.

## 465 4 Conclusions

In the present study, we investigated plant species as sources of SFAs in atmospheric aerosols based on the mass size distributions of aerosol SFAs and leaf samples simultaneously obtained from a cool-temperate TMK forest site. Among the five SFAs identified, *n*-nonacosan-10-ol was the most abundant in the aerosols. The concentration of *n*-nonacosan-10-ol in the bulk aerosol was highest in spring, followed by autumn. Across all the seasons, the size range of most SFAs coincided with

https://doi.org/10.5194/egusphere-2025-4483 Preprint. Discussion started: 22 October 2025

© Author(s) 2025. CC BY 4.0 License.

EGUsphere Preprint repository

the supermicrometer ( $D_p > 1~\mu m$ ) range, which accounted for 78% of the total aerosol mass. In spring and summer, the mass

size distribution of *n*-nonacosan-10-ol exhibited a peak diameter of  $> 7.2 \mu m$ . On average, the peak size of *n*-nonacosan-10-ol

mass shifted to smaller ranges as the seasons progressed. This seasonal change in the size distribution of aerosol SFAs was

similar to that observed at other forest sites in Sapporo (Cui et al., 2023).

Among the leaf samples of various plant species collected at the study site, *n*-nonacosan-10-ol was detected in the leaf samples

of two coniferous evergreen species, Sakhalin fir (Abies sachalinensis) and Sakhalin spruce (Picea glehnii). In contrast, SFAs

were not detected in the broad leaves of deciduous tree species. This result was supported by a previous study that showed that

n-nonacosan-10-ol is generally found in the leaf waxes of conifers. Additionally, most of the n-nonacosan-10-ol (97%) in the

conifer leaf samples was found in the free (non-esterified) form, which was consistent with previous findings reported in the

literature.

480 The mass of *n*-nonacosan-10-ol per needle leaf was largest in spring, followed by autumn. This seasonal trend was attributed

to the significant production and accumulation of plant leaf wax in spring and autumn, similar to that of aerosols. The

comparison between SFAs in the aerosol and leaf samples and meteorological parameters suggested that the atmospheric

emission strength of n-nonacosan-10-ol from the leaf surface was influenced by the phenological stage of the coniferous leaves

and local wind speeds, particularly in spring. Overall, the results suggest that n-nonacosan-10-ol, the major SFA compound,

can be used as a tracer of PBAP originating from coniferous leaf wax. Considering the global coverage of the biomass of

coniferous trees, their wax should be considered as a potential source of PBAPs, which can reduce uncertainties in estimating

the atmospheric emission flux of PBAP.

Data availability

The measurement data of the samples are provided in the Supplementary Material. All the other data are available upon request.

490 Author contributions

YC and YM designed the study and prepared the manuscript. YC and ET measured SFAs and relevant parameters. YC and

YM sampled aerosols and plant leaves. YC, YM, and ET analyzed the data.

**Competing interests** 

The authors declare that they have no conflict of interest.

# 495 Acknowledgements

We thank Masahiro Nakamura and the staff of the TMK experimental forest site for managing the sampling site and for their help with the initial setup for aerosol and leaf sampling.

## **Financial supports**

520

This research was supported by JST SPRING (grant number: JPMJSP2119) via the DX Doctoral Fellowship of Hokkaido University.

- Angst, G., Cajthaml, T., Angst, Š., Mueller, K. E., Kögel-Knabner, I., Beggel, S., Kriegs, S., and Mueller, C. W.: Performance of base hydrolysis methods in extracting bound lipids from plant material, soils, and sediments, *Org. Geochem.*, 113, 97-104, https://doi.org/10.1016/j.orggeochem.2017.08.004, 2017.
  - Caffarra, A. and Donnelly, A.: The ecological significance of phenology in four different tree species: effects of light and temperature on bud burst, *Int J Biometeorol*, 55, 711-721, 10.1007/s00484-010-0386-1, 2011.
  - CAPE, J. N. and PERCY, K. E.: Environmental influences on the development of spruce needle cuticles, *New Phytol.*, 125, 787-799, https://doi.org/10.1111/j.1469-8137.1993.tb03928.x, 1993.
    - Chen, J., Kawamura, K., Hu, W., Liu, C.-Q., Zhang, Q., and Fu, P.: Terrestrial lipid biomarkers in marine aerosols over the western North Pacific during 1990–1993 and 2006–2009, *Sci. Total Environ.*, 797, 149115, https://doi.org/10.1016/j.scitotenv.2021.149115, 2021.
- Cui, Y., Tachibana, E., Kawamura, K., and Miyazaki, Y.: Origin of secondary fatty alcohols in atmospheric aerosols in a cooltemperate forest based on their mass size distributions, *Biogeosciences Discuss.*, 2023, 1-17, 10.5194/bg-2023-73, 2023.
  - Després, V., Huffman, J. A., Burrows, S. M., Hoose, C., Safatov, A., Buryak, G., Fröhlich-Nowoisky, J., Elbert, W., Andreae, M., Pöschl, U., and Jaenicke, R.: Primary biological aerosol particles in the atmosphere: a review, *Tellus B: Chemical and Physical Meteorology*, 64, 15598, 10.3402/tellusb.v64i0.15598, 2012.
  - Dommisse, A., Wirtz, J., Koch, K., Barthlott, W., and Kolter, T.: Synthesis of (S)-Nonacosan-10-ol, the Major Component of Tubular Plant Wax Crystals, *Eur. J. Org. Chem.*, 2007, 3508-3511, https://doi.org/10.1002/ejoc.200700262, 2007.
  - Fu, P., Kawamura, K., Chen, J., and Barrie, L. A.: Isoprene, monoterpene, and sesquiterpene oxidation products in the high Arctic aerosols during late winter to early summer, *Environ Sci Technol*, 43, 4022-4028, 10.1021/es803669a, 2009.
  - Gagosian, R. B., Peltzer, E. T., and Merrill, J. T.: Long-range transport of terrestrially derived lipids in aerosols from the south Pacific, *Nature*, 325, 800-803, 10.1038/325800a0, 1987.
- Hader, J. D., Wright, T. P., and Petters, M. D.: Contribution of pollen to atmospheric ice nuclei concentrations, *Atmos. Chem. Phys.*, 14, 5433-5449, 10.5194/acp-14-5433-2014, 2014.

550

- Hiura, T.: Stochasticity of species assemblage of canopy trees and understorey plants in a temperate secondary forest created by major disturbances, *Ecol. Res.*, 16, 887-893, 10.1046/j.1440-1703.2001.00449.x, 2001.
- Hiura, T.: Estimation of aboveground biomass and net biomass increment in a cool temperate forest on a landscape scale,

  Forest Ecosystems and Environments: Scaling Up from Shoot Module to Watershed, 31-37, 2005.
- Huth, M. A., Huth, A., and Koch, K.: Self-assembly of Eucalyptus gunnii wax tubules and pure β-diketone on HOPG and glass, *Beilstein J. Nanotechnol.*, 12, 939-949, https://doi.org/10.3762/bjnano.12.70, 2021.
  - Jenks, M., Gaston, C., Goodwin, M., Keith, J., Teusink, R., and Wood, K.: Seasonal variation in cuticular waxes on Hosta genotypes differing in leaf surface glaucousness, *HortScience*, 37, 673–677, 2002.
- Koch, K., Dommisse, A., and Barthlott, W.: Chemistry and crystal growth of plant wax tubules of lotus (Nelumbo n ucifera) and nasturtium (Tropaeolum m ajus) leaves on technical substrates, *Cryst. Growth Des.*, 6, 2571-2578, 2006.
  - Koch, K., Dommisse, A., Niemietz, A., Barthlott, W., and Wandelt, K.: Nanostructure of epicuticular plant waxes: Self-assembly of wax tubules, *Surface Science*, 603, 1961-1968, https://doi.org/10.1016/j.susc.2009.03.019, 2009.
- Koivisto, E., Ladommatos, N., and Gold, M.: Systematic study of the effect of the hydroxyl functional group in alcohol molecules on compression ignition and exhaust gas emissions, *Fuel*, 153, 650-663, https://doi.org/10.1016/j.fuel.2015.03.042, 2015.
  - Kolattukudy, P., Espelie, K., and Rowe, J.: Natural Products of Woody Plants, by JW Rowe, Springer-Verlag, Berlin, 304, 1989.
  - Kolattukudy, P. E.: Chemistry and Biochemistry of Natural Waxes, Elsevier Scientific Publishing Company1976.
- Kreyling, J., Wiesenberg, G. L. B., Thiel, D., Wohlfart, C., Huber, G., Walter, J., Jentsch, A., Konnert, M., and Beierkuhnlein, C.: Cold hardiness of Pinus nigra Arnold as influenced by geographic origin, warming, and extreme summer drought, *Environ. Exp. Bot.*, 78, 99-108, https://doi.org/10.1016/j.envexpbot.2011.12.026, 2012.
  - Laila, R., Robin, A. H. K., Yang, K., Park, J.-I., Suh, M. C., Kim, J., and Nou, I.-S.: Developmental and Genotypic Variation in Leaf Wax Content and Composition, and in Expression of Wax Biosynthetic Genes in Brassica oleracea var. capitata, *Frontiers in Plant Science*, 7, 10.3389/fpls.2016.01972, 2017.
  - Liu, Z., Hikosaka, K., Li, F., and Jin, G.: Variations in leaf economics spectrum traits for an evergreen coniferous species: Tree size dominates over environment factors, *Funct. Ecol.*, 34, 458-467, https://doi.org/10.1111/1365-2435.13498, 2020.
  - Lukas, M., Schwidetzky, R., Kunert, A. T., Pöschl, U., Fröhlich-Nowoisky, J., Bonn, M., and Meister, K.: Electrostatic Interactions Control the Functionality of Bacterial Ice Nucleators, *Journal of the American Chemical Society*, 142, 6842-6846, 10.1021/jacs.9b13069, 2020.
  - Matas, A. J., Sanz, M., x, a, J., and Heredia, A.: Studies on the structure of the plant wax nonacosan-10-ol, the main component of epicuticular wax conifers, *Int. J. Biol. Macromol.*, 33, 31-35, https://doi.org/10.1016/S0141-8130(03)00061-8, 2003.
  - Miyazaki, Y., Gowda, D., Tachibana, E., Takahashi, Y., and Hiura, T.: Identification of secondary fatty alcohols in atmospheric aerosols in temperate forests, *Biogeosciences*, 16, 2181-2188, 10.5194/bg-16-2181-2019, 2019.

- Mohammadian, M. A., Watling, J. R., and Hill, R. S.: The impact of epicuticular wax on gas-exchange and photoinhibition in Leucadendron lanigerum (Proteaceae), *Acta Oecol.*, 31, 93-101, https://doi.org/10.1016/j.actao.2006.10.005, 2007.
  - Nakai, Y., Kitamura, K., Suzuki, S., and Abe, S.: Year-long carbon dioxide exchange above a broadleaf deciduous forest in Sapporo, Northern Japan, *Tellus B: Chemical and Physical Meteorology*, 55, 305-312, 2003.
- Nikolić, B., Todosijević, M., Đorđević, I., Stanković, J., Mitić, Z. S., Tešević, V., and Marin, P. D.: Nonacosan-10-ol and n-565 Alkanes in Leaves of Pinus pinaster, *Natural Product Communications*, 15, 1934578X20926073, 2020.
- O'Sullivan, D., Murray, B. J., Ross, J. F., Whale, T. F., Price, H. C., Atkinson, J. D., Umo, N. S., and Webb, M. E.: The relevance of nanoscale biological fragments for ice nucleation in clouds, *Sci. Rep.*, 5, 8082, 10.1038/srep08082, 2015.
  - Pan, Y., Birdsey, R. A., Fang, J., Houghton, R., Kauppi, P. E., Kurz, W. A., Phillips, O. L., Shvidenko, A., Lewis, S. L.,
- Canadell, J. G., Ciais, P., Jackson, R. B., Pacala, S. W., McGuire, A. D., Piao, S., Rautiainen, A., Sitch, S., and Hayes, D.: A Large and Persistent Carbon Sink in the World's Forests, *Science*, 333, 988-993, doi:10.1126/science.1201609, 2011.
- Post-Beittenmiller, D.: BIOCHEMISTRY AND MOLECULAR BIOLOGY OF WAX PRODUCTION IN PLANTS, *Annu. Rev. Plant Biol.*, 47, 405-430, https://doi.org/10.1146/annurev.arplant.47.1.405, 1996.
  - Qiu, Y., Odendahl, N., Hudait, A., Mason, R., Bertram, A. K., Paesani, F., DeMott, P. J., and Molinero, V.: Ice nucleation efficiency of hydroxylated organic surfaces is controlled by their structural fluctuations and mismatch to ice, *Journal of the American Chemical Society*, 139, 3052-3064, 2017.
  - Rathnayake, C. M., Metwali, N., Jayarathne, T., Kettler, J., Huang, Y., Thorne, P. S., O'Shaughnessy, P. T., and Stone, E. A.: Influence of rain on the abundance of bioaerosols in fine and coarse particles, *Atmos. Chem. Phys.*, 17, 2459-2475, 10.5194/acp-17-2459-2017, 2017.
- Shepherd, T. and Wynne Griffiths, D.: The effects of stress on plant cuticular waxes, *New Phytol.*, 171, 469-499, https://doi.org/10.1111/j.1469-8137.2006.01826.x, 2006.
  - Simões, R., Miranda, I., and Pereira, H.: Effect of Seasonal Variation on Leaf Cuticular Waxes' Composition in the Mediterranean Cork Oak (Quercus suber L.), *Forests*, 13, 1236, 2022.
  - Simoneit, B. R. T., Kobayashi, M., Mochida, M., Kawamura, K., Lee, M., Lim, H.-J., Turpin, B. J., and Komazaki, Y.: Composition and major sources of organic compounds of aerosol particulate matter sampled during the ACE-Asia campaign, *Journal of Geophysical Research: Atmospheres*, 109, https://doi.org/10.1029/2004JD004598, 2004.
- Takahashi, Y., Saigusa, N., Hirata, R., Ide, R., Fujinuma, Y., Okano, T., and Arase, T.: Characteristics of temporal variations in ecosystem CO<sub>2</sub> exchange in a temperate deciduous needle-leaf forest in the foothills of a high mountain, *Journal of Agricultural Meteorology*, 71, 302-317, 10.2480/agrmet.D-14-00009, 2015.
- Tegen, I. and Schepanski, K.: Climate Feedback on Aerosol Emission and Atmospheric Concentrations, *Current Climate Change Reports*, 4, 1-10, 10.1007/s40641-018-0086-1, 2018.
  - Tobo, Y., Prenni, A. J., DeMott, P. J., Huffman, J. A., McCluskey, C. S., Tian, G., Pöhlker, C., Pöschl, U., and Kreidenweis, S. M.: Biological aerosol particles as a key determinant of ice nuclei populations in a forest ecosystem, *Journal of Geophysical Research: Atmospheres*, 118, 10,100-110,110, https://doi.org/10.1002/jgrd.50801, 2013.

- Tomaszewski, D.: The wax layer and its morphological variability in four European Salix species, *Flora Morphology,* 595 *Distribution, Functional Ecology of Plants*, 199, 320-326, https://doi.org/10.1078/0367-2530-00159, 2004.
  - Vazquez de Vasquez, M., Carter-Fenk, K., Beasley, E., Simpson, J., Allen, H., and McCaslin, L.: Ice nucleation insights: interfacial electric fields and fatty alcohol and acid hydration, AGU Fall Meeting Abstracts, A035-0009,
    - Wang, J., Xu, Y., Zhou, L., Shi, M., Axia, E., Jia, Y., Chen, Z., Li, J., and Wang, G.: Disentangling temperature effects on leaf wax n-alkane traits and carbon isotopic composition from phylogeny and precipitation, *Org. Geochem.*, 126, 13-22, https://doi.org/10.1016/j.orggeochem.2018.10.008, 2018.
    - Yu, L. E., Shulman, M. L., Kopperud, R., and Hildemann, L. M.: Characterization of organic compounds collected during southeastern aerosol and visibility study: water-soluble organic species, *Environ. Sci. Technol.*, 39, 707-715, 2005.
    - Yuan, Z., Shi, X., Jiao, F., and Han, F.: N and P resorption as functions of the needle age class in two conifer trees, *Journal of Plant Ecology*, 11, 780-788, 10.1093/jpe/rtx055, 2018.
- 605 Zhang, Z., Mistry, D., and Jetter, R.: Micromorphological and chemical characterization of Drimys winteri leaf surfaces: The secondary alcohols forming epicuticular wax crystals are accompanied by alkanediol, alkanetriol and ketol derivatives, *Plant Cell Physiol.*, 65, 1245-1260, 2024.