# Peer review of "Conifer leaf wax acts as a source of secondary fatty alcohols in atmospheric aerosols"

_EGUsphere, 2025_

## Author Comment (AC1)

**Responses to the comments of Referee#1**

General comments:

Understanding the emission of PBAPs into the atmosphere and their impact is of high importance to the scientific community to deepen the understanding of the complex dynamics of our climate. This study focuses on the emission of secondary fatty acids as PBAPs tracers. The authors identified certain coniferous trees in Japanese forests and investigate SFAs from these trees by first directly assessing SFA concentration from the trees surface and compare these to SFAs in the atmosphere by sampling aerosol in proximity. Moreover, they set this into a seasonal perspective and compare it to a small number of meteorological variables.

The methodology is well thought out and the basis for good scientific work. The dataset is rather small but valuable, and makes the authors conclude with seasonal trends of SFA emission and that coniferous trees are the main emitter of SFAs. While the methodology provides a good basis for scientific quality, to this reviewer it was hard to follow the story. The results are presented step by step, however sometimes the reason why something is presented (or not) is inconclusive. The fact that winter is missing for most of the seasonal data is a bummer, also the fact that for summer there is only one datapoint. To me the question arises how to conclude with seasonal trends if only two seasons are supported with somewhat reliable data. Regardless of the reason for the missing data, the manuscript must acknowledge this limitation in the abstract and conclusion, and refrain from generalizing to seasonal trends, when it is rather a spring-autumn comparison.

Still, this manuscript provides interesting data worth publishing, but I suggest a major revision of the storyline and the data presentation, which will be described below.

**Reply: We are grateful for the referee's thorough assessment and the constructive suggestion to improve the manuscript. We have revised the manuscript by taking account of the comments.**

Major points:

- Seasonal trends: With field measurements it can be hard to get data for all seasons, however it is not well described why there is only one week of data in the summer and no data in the winter for the aerosol data. For leaves, Figure 7 presents winter data from a different species (Sakhalin spruce) than the other seasons (Sakhalin fir). This inconsistency makes the comparison invalid, otherwise clearly state why it is valid.

The manuscript could be reframed as a comparative study of Spring vs. Autumn. The Abstract and Conclusion must be revised to remove broad generalizations about "seasonal variations" where data is insufficient. I strongly suggest removing the winter data point from Figure 7.

**Reply: There are two major reasons for the limited or no data in summer and winter:**

1) **Our previous studies reported that SFAs in aerosols showed increased concentrations in summer and autumn at several different forest sites (Miyazaki et al., 2019; Cui et al., 2023). Based on these studies, we focused on those two seasons to elucidate the source of SFAs in this study.**
2) **The lack of aerosol data in winter was due to accessibility to the sampling site. Specifically, snow accumulation prevented us from accessing the sampler on the tower in winter. Instead, we could obtain leaf samples of *Sakhalin spruce* similar to *Sakhalin fir* at location about 100 m away from the tower. This is why we show the data of *Sakhalin spruce* only for winter just for reference.**

**According to the comment, we have revised the manuscript on the points below:**

1. **The winter data originally shown in Figure 7 has been removed. As pointed out by the referee, this change makes the comparison stricter.**
2. **We have revised the statement to focus on the comparison of data between spring and autumn as suggested. Specifically, we have revised the Abstract and Conclusion to state the limited data not to generalize the seasonal trends of SFAs as follows:**

**Abstract (L.11):** *"In this study, we collected size-segregated aerosols and leaf samples from various plant species at a cool-temperate forest site in Hokkaido, northern Japan, mainly in spring and autumn."*

**L.15:** *"Despite the limited data, the mass of n-nonacosan-10-ol per leaf exhibited a seasonal difference similar to that of the aerosol SFA concentrations between spring and autumn."*

**Conclusion (L.460):** *"For all the samples, the size range of most SFAs coincided with the supermicrometer ($D_p > 1 \mu m$) range, which accounted for 78% of the total aerosol mass. In particular, the mass size distribution of n-nonacosan-10-ol exhibited a peak diameter of $> 7.2 \mu m$ in spring. On average, the peak size of n-nonacosan-10-ol mass shifted to smaller ranges in autumn compared to spring. This seasonal difference…"*

- Discrepancy Between Aerosol and Leaf Trends:

  o Aerosols (Figure 4): Spring concentrations are vastly higher than Autumn. Summer and Autumn are effectively indistinguishable given the large error bars in Autumn and the lack of variance data for Summer.

  o Leafs (Figure 7): Spring and Autumn masses are nearly identical (the authors even state the difference is "insignificant").

  o You state the trends in conifer leafs are similar as in the aerosol samples in your conclusion. Considering the previous comment the trends are not "similar", rather contradicting. This needs to be addressed.

**Reply:**

**As the referee pointed out, the amount of SFA in the plant leaves were similar in the two seasons. What we want to mention about the seasonal difference is that the concentrations (mass) of SFAs in spring and autumn were higher relative to those in summer, which was observed both in aerosol and leaf samples. Therefore, it is not contradictory, while we agree that the original statement might cause misunderstanding. Taking account of the comment, we have revised the statement as follows:**

**L.272:** *The trend of higher mass of n-nonacosan-10-ol in the conifer leaf samples in spring and autumn relative to summer was similar to that of the mass concentration in aerosols (Fig. 4).*

- SFA as PBAP tracer: A central argument of the paper is that SFAs can serve as tracers for bulk water-insoluble organic carbon (WIOC). However, the evidence for this is relegated to Figure S2 in the Supplement. Since this relationship is foundational to the paper's significance, this figure should be moved to the main manuscript.

**Reply: We agree that the correlation between SFAs and WIOC is foundational to the significance of our study. According to the comment, Figure S2 has been moved to the main text as Figure 12.**

Minor points:

- Introduction: The link between SFAs and ice nucleation is presented somewhat tenuously. The cited study (Qiu et al.) is simulation-based. I suggest citing experimental studies to strengthen this motivation (e.g. https://pubs.rsc.org/en/content/articlehtml/2024/ea/d4ea00066h).

**Reply: According to the comment, we have included the suggested reference in the Introduction section to strengthen the motivation of our study as follows:**

**L.27:** *"Qiu et al. (2017) suggested that monolayers of n-alkyl alcohols with carbon numbers up to 30 can act as efficient ice nucleants based on molecular simulations. This potential has been supported by experimental evidence, which showed that fatty alcohol particles have significant ice nucleation ability depending on the carbon chain length (Mehndiratta et al., 2024). This ice nucleation ability is a common characteristic of long-chain fatty alcohols (FAs) (Vazquez De Vasquez et al., 2020)."*

- Section 2.1 (Sampling Gaps): Please provide a brief explanation for the lack of winter aerosol data and the limited summer sampling. While logistical challenges are common, transparency is required.

**Reply: The lack of winter data was due to accessibility to the sampling site. Specifically, snow accumulation exceeding 1-meter height prevented us from accessing the sampling site in winter. The limited sampling in summer was our strategic decision based on our previous studies (Miyazaki et al., 2019; Cui et al., 2023), which showed that concentrations of secondary fatty alcohol (SFA) increased in spring and autumn with decreasing values in summer as well as in winter. Therefore, our study focused on SFAs in spring and autumn when their concentrations in aerosols show increase to elucidate the sources.**

**According to the comment, we added a brief explanation about the points above in Section 2.1 as follows:**

**L. 100: "*The lack of data in winter was due to heavy snow accumulation exceeding 1-m height around the sampling site.*"**

- Section 2.3 (Filter Analysis): Why do the filter cut areas differ between the bottom stage and upper stages? Furthermore, impactors often deposit particles non-uniformly (center-line concentration). Please clarify how the cuts were taken to ensure they were representative of the total loading.

**Reply: First of all, the sizes, shapes, and structures of the impactors (and thus those of sample filters) are completely different between one bottom stage and four upper stages. For the upper stages, particles are collected on each filter with multiple parallel slots. The outside dimensions of the plates in each stage are 6 inches × 7 inches. For the measurement, we cut one piece from a filter corresponding to one slot per stage of the upper plates. On the other hand, the bottom stage collects particles on a filter (without slots) of 8 inches × 10 inches, where we punched a filter with a diameter of 22 mm. Because it is difficult to cut exactly the same filter area from each stage, the area cut was set to be approximately 2–3 cm$^2$ in each stage.**

**As the referee pointed out, deposited particles onto filters via impactors are often not uniform. We cut the filter from the center as much as possible. In fact, we made multiple analysis of filter cut from the center and edge of one filter to find that the difference in the aerosol carbon mass at the different parts was within 6%.**

**We made additional descriptions on the points above as follows:**

**L. 117: *"Uniformity of the particle deposition onto the filter was evaluated by analyzing filters cut from the center and edge of one filter to find that the difference in the collected aerosol mass between the two different parts was less than 6%."***

- Section 2.5 (Sample Processing Bias): The manuscript states that broadleaf samples were ground/homogenized, while coniferous needles were not. Please elaborate on this decision.

**Reply: The original statement did not provide enough explanation on the treatment of the leaf samples. We initially extracted the broadleaf samples without grinding, the method of which is the same as the coniferous needle leaves. However, no SFAs were detected in those broadleaf samples. To make sure of the absence of SFAs rather than inefficient extraction from the leaves, we ground the broadleaves to increase the possibility of extraction. Even with this method, SFAs remained below the detection limit. In contrast, SFAs in the coniferous needle leaves were detect by the initial extraction method without any grinding. In the revised manuscript, we have corrected the statement to clarify the method as follows:**

**L. 153: *"Initially, both broadleaf and coniferous leaf samples were extracted without grinding. If SFAs were not detected, the corresponding samples were then ground and homogenized in a mortar to try to detect SFAs again."***

- Section 3.2 / Table 1 (Detection Limits): "Not Detected" (ND) is used for broadleaves. Please state the Limit of Detection (LOD) for the analytical method.

**Reply: In this study, the limit of detection (LOD) is defined as the concentration with the signal-to-noise ratio (S/N) less than 3 (i.e., no distinct peak was observed), which is determined to be 0.01 ng m⁻³. Concentration values below this LOD are reported as ND. We have made additional statement on this LOD in Section 3.2 and the caption of Table 1 as follows:**

**Section 3.2 (L.232):** *"In this study, the limit of detection (LOD) was defined as the concentration with a signal-to-noise ratio (S/N) of less than 3, which was determined to be 0.01 ng m⁻³."*

**Caption of Table 1:** *"ND indicates that a compound was not detected, whose concentration was below the lower limit of detection (LOD; 0.01 ng m⁻³)."*

- Figure 10 (Meteorology): The correlation between wind speed and aerosol concentration in Spring appears weak visually. The authors should provide a scatter plot or statistical correlation coefficient to substantiate the claim that wind drives emissions. The inverse relationship in Autumn further complicates this hypothesis.

**Reply: One of the reasons for visually unclear correlation was that aerosol concentration in Figure 10a was shown in log scale, where the difference in the concentrations among the three samples was not visually apparent. Due to the limited number of samples (N=3 in each season), a statistical correlation analysis is not robust. Therefore, we have not provided a scatter plot. Instead, we have revised Figure 10a to be shown in linear scale so that the correlation between aerosol concentration and local wind speeds becomes clearer than the original figure. For the autumn samples, the relationship between aerosols and meteorological parameters was not clear as we described in the original text.**

- Sugar Compounds: The methodology mentions measuring sugar compounds, but these results do not appear to be discussed. Please either remove the mention or include the data.

**Reply: Originally, we measured sugar compounds to investigate their relations with SFAs. However, the sugar compounds did not show clear relationships with SFAs in this study. To appropriately focus on the SFAs discussed in this study, we have deleted the terms "...and sugar compounds" in the title of Section 2.3 and**

**"...and related compounds"** in the first line of that subsection, according to the referee's comment.

Technical points:

- Figure 1: A photo of the sampling site/equipment would be helpful for context.

**Reply: A photo of the sampling site has been added to Figure 1.**

- Figure 3: The legend is disproportionately large; please resize.

**Reply: The legend has been resized as suggested.**

- Line 90: "temperature humidity" should be "temperature and humidity."

**Reply: Corrected as pointed out (L.93).**

- Line 95: The sentence "Consequently, the present study emphasizes..." lacks context. Please clarify that this refers to the data availability of Spring and Autumn.

**Reply: We have revised the statement in this section to explicitly state that the study focuses on the data in spring and autumn, as follows:**

**L. 96: *"Our previous studies showed that mass concentrations of SFAs in forest aerosols increased in spring and autumn (Miyazaki et al., 2019; Cui et al., 2023). Consequently, the present study particularly focused on these two seasons."***

- Line 213-214: The claim of a peak in the 1.5–3.0 µm range is not well supported due to the large error bars; the peak definition is ambiguous in this range.

**Reply: What we want to mention is that the peak diameter > 7.2 µm was not evident in autumn and we are not particular about the specific range of 1.5–3.0**

**µm. We have revised the statement regarding the peak in the 1.5–3.0 µm range as follows:**

**L. 215:** *"In spring, the peak diameter of n-nonacosan-10-ol was larger than 7.2 µm. On the other hand, such peak was not evident in autumn, when the peak was observed in rather smaller size range."*

- Line 290: Please define the specific months considered "growing seasons" for these specific tree species.

**Reply: In the revised manuscript, we have defined growing seasons as a period from March to May (L.44).**

- Lines 315-327: The detailed description of biosynthesis pathways seems tangential to the study's focus on emission fluxes. Consider shortening this section.

**Reply: According to the comment, we have deleted most of the description on the biosynthesis pathway to be shortened in the revised manuscript (L.312–316).**

---

## Author Comment (AC2)

**Responses to the comments of Referee#2**

General comments:

This manuscript, which builds on a previous study by Cui et al. (2023), examines the origin and emission of secondary fatty alcohols (SFAs) in size-segregated atmospheric aerosols collected at a cool-temperate forest site in Hokkaido, Japan. The authors identified $n$-nonacosan-10-ol as the predominant SFA produced by coniferous trees and compared the concentrations in conifer leaves to levels in aerosol samples collected across seasons. The authors observed a seasonal variability, but due to extremely limited sampling in the summer and winter, it may be more accurate to consider their findings a comparison of spring and autumn $n$-nonacosan-10-ol levels. However, the methods are comprehensive, and overall, the study provides interesting new insights into a biogenic source of atmospheric aerosols, despite a somewhat small dataset. Therefore, I support the publication of this manuscript in BG, after addressing the following comments.

**Reply: We sincerely thank the referee for the constructive comments on our work. We have carefully addressed all the comments and revised the manuscript accordingly.**

Specific Comments

1. Fig 7/Table S4: The mass of $n$-nonacosan-10-ol shown for winter in Fig. 7 does not match the data presented in Table S4. According to Fig. 7, the mass of $n$-nonacosan-10-ol per leaf in winter is 2.32±34 mg, but the average mass of the winter values shown in Table S4 can be calculated as 16.17 mg. In addition, the winter $n$-nonacosan-10-ol masses are exactly the same as those for summer (16.9, 14.3, and 17.3 mg), and two of the three leaf weights shown for summer and winter are identical (4.51 and 4.58 mg). Please double check the data shown in Table S4 and confirm that the values align with what is shown in Fig. 7.

**Reply 1: We apologize for the wrong numbers shown in the original Table S4. The values in winter originally shown in Table S4 were mistakenly presented as the referee pointed out, while the data presented in Figure 7 is correct. We have now corrected the values in the revised Table S4.**

2. Table S3: Does 'deep yellow' refer to the brown part of the leaf? I might have missed it, but it seems that this specific color is not defined in the text.

**Reply 2: Yes, the "deep yellow" in Table S3 is identical to the "brown" part described in the main text, which we made a mistake to specify. We have corrected the term "deep yellow" to "brown" in Table S3 to be identical to the description in the text.**

Technical Corrections

1. Line 104 is missing the word *the* ('These species were selected because they dominate *the* study site in the forest').

**Reply 3: Corrected.**

2. Table S1 should be referenced in the main text.

**Reply 4: Table S1 has been referenced in Line 187 in the revised manuscript.**

3. Table S2 should be referenced in the caption of Fig. 6 or in nearby text.

**Reply 5: According to the comment, we have referred Table S2 in the caption of Figure 6.**

4. A reference to Table S3 should be included in the main text.

**Reply 6: Table S3 has been refereed in the captions of Figures 8 and 9.**

5. Line 457 and 457: The supplement is referred to as Supplementary Material and Supplementary Information, respectively. Please select one for consistency.

**Reply 7: Because the original Fig. S2 has been moved to Figure 12, the terms "Supplementary ..." have been deleted in the main text.**

6. Lines 476 – 479 need references.

**Reply 8: We added the following references to the text.**

**Matas, A. J., Sanz, M., x, a, J., and Heredia, A.: Studies on the structure of the plant wax nonacosan-10-ol, the main component of epicuticular wax conifers, Int. J. Biol. Macromol., 33, 31-35, https://doi.org/10.1016/S0141-8130(03)00061-8, 2003.**

**Kolattukudy, P., Espelie, K., and Rowe, J.: Natural Products of Woody Plants,** *by JW Rowe, Springer-Verlag, Berlin*, **304, 1989.**

**Kolattukudy, P. E.: Chemistry and Biochemistry of Natural Waxes, Elsevier Scientific Publishing Company, 1976.**